# Marine Collision Avoidance Route Planning Model for MASS Based on Domain-Based Predicted Area of Danger

Chao-Wei Lu [1,2,3,4], Chao-Kuang Hsueh [4], Yung-Lin Chuang [4], Ching-Ming Lai [1,3,5,*] and Fuh-Shyong Yang [3]

1   Intelligent Transportation Development Center, National Chung Hsing University, No. 145, Xingda Rd., South District, Taichung City 402, Taiwan; clu26@sheffield.ac.uk
2   Management School, The University of Sheffield, Conduit Road, Sheffield S10 1FL, UK
3   Institute of Transportation and Communication Management Science, National Cheng Kung University, No. 1, Daxue Rd., East District, Tainan City 701, Taiwan; r58101033@gs.ncku.edu.tw
4   Department of Merchant Marine, National Taiwan Ocean University, No. 2, Beining Rd., Zhongzheng District, Keelung City 20224, Taiwan; ckhsueh@mail.ntou.edu.tw (C.-K.H.); julie31428@gmail.com (Y.-L.C.)
5   Department of Electrical Engineering, National Chung Hsing University, No. 250, Guoguang Rd., South District, Taichung City 40227, Taiwan
*   Correspondence: pecmlai@gmail.com

**Abstract:** When the own ship encounters target ships in a close-quarter situation, an officer on watch needs to safely and timely alter the course of the vessel to avoid a collision. If ECDIS can automatically collect the navigation parameters and plot areas of collision as quasi-static obstruction areas, it will be much easier for seafarers to implement effective route planning. Hence, this study focuses on developing the MCARP model as a theoretical concept based on the DPAD model and LCD model. By operating the MCARP using ArcGIS, DPADs and several effective collision avoidance routes can be generated and imported into ECDIS based on AIS information at large scales and ample time. The graphic overlay of DPADs and effective routes on ECDIS can serve as a collision avoidance strategy reference for the personnel controlling maritime autonomous surface ships. Finally, different ships encountering situations were input into a Transas navigational simulator. The simulation results showed that own ship could avoid collisions with multiple target ships at distances larger than the preset collision avoidance distances, which also indicated that MCARP is practically feasible.

**Keywords:** ArcGIS; MASS; domain-based predicted area of danger; route planning; ship domain

## 1. Introduction

Since the era of merchant ship enlargement, ship speed maximization, and cargo containerization, ship collision avoidance has been one of the most concerning topics for the shipping industry. According to shipping accident statistics of the European Maritime Safety Agency (EMSA), there were 2637 maritime incidents reported in the territorial waters and inland waterways of European Union countries, and colliding incidents account for 43.0% of the figures [1]. Human errors account for between 60% and 90% of the causes of maritime incidents [2], and the corresponding figures vary according to the vessel type. Nautical instruments have been adapted to modern advanced technology levels, becoming essential and irreplaceable equipment to achieve ship collision avoidance. Seafarers are obliged to be equipped with impeccable skillsets for operating the corresponding devices, which are accumulated with hours of watchkeeping experience.

In the modern practice of marine navigation, the safety of navigation is mainly assured using frequent operation and monitoring of automatic radar plotting aids (ARPA) and the Electronic Chart Display and Information System (ECDIS). ARPA obtains collision parameters such as the closest point of approach (CPA) and time to closest point of approach (TCPA) by emitting electromagnetic waves at objects, as well as simulating collision avoidance scenarios for the future five minutes [3]. The ECDIS inputs require navigation

parameters from the Global Positioning System (GPS), Automatic Identification System (AIS), ARPA, and Gyro Compass [4]; thus, transforming the data into collision avoidance references on an electronic navigation chart (ENC) is necessary for the officer on watch (OOW) to formulate decision-making strategies.

However, collision avoidance decision-making can sometimes be relatively intricate for the OOW in multiple-ship encountering situations within high-density maritime traffic areas such as the Malacca Strait [5]. Although ARPA and the ECDIS have become the main decision support systems for ship collision avoidance in navigation practice, an OOW could mostly rely on their own watchkeeping experiences to alter the waypoint while keeping a sharp lookout on nearby waters [6]. The non-existence of automatic and effective collision avoidance route references from ARPA and the ECDIS might impose potential defects on collision avoidance strategies, which leads to serious maritime accidents [7]. Therefore, an automation module on the ECDIS that combines the concept of collision danger areas and effective collision avoidance route planning based on verified AIS information would considerably benefit the accuracy of collision avoidance strategies made by an OOW [8].

Automation is one of the crucial technologies in the maritime industry. For instance, the unmanned ship, automatic fender system, automatic gantry crane, and automatic container truck trailer are essential elements of the future trend in intelligent ports. Within the international intelligent maritime system, artificial intelligence (AI) plays a significant role in marine automation, and it will likely replace the OOW for making collision avoidance decisions in merchant marines [9]. In fact, due to impacts on the safety, business, and operational environment of autonomous ships, the International Maritime Organization (IMO) developed the Maritime Autonomous Surface Ships (MASS) code as a non-mandatory regulation. MASS can be divided into four degrees. From degree 1 to degree 4, an automation system with gradually evolved levels is equipped on ships; details of the four degrees of automation on ships are presented in Table 1, which were proposed in [10].

**Table 1.** Details of the four degrees of automation on ships [10].

| Degree Level | Level of Automation | OOW Intervention | Role of the OOW |
| --- | --- | --- | --- |
| 1 | Automated processes | Yes | Supervision and operation |
| 2 | Remote controlling with seafarers | Yes | Supervision and remote controlling |
| 3 | Remote controlling without seafarers | No | Remote controlling and monitoring |
| 4 | Fully autonomous | No | Remote monitoring and emergency management |

Following the automation trend in the maritime industry, automatic collision avoidance route planning models should be installed on all four degrees of MASS, as this will stimulate the development speed of fully autonomous ships. Furthermore, regarding the operational accuracy of fully autonomous ships, experimental results have shown that collision avoidance routes made by fully autonomous ships are likely to be human-level [11]. Autonomous ships could also be more efficient and environmentally friendly when compared to current merchant ships in the maritime system, which can help the global naval industry reach the IMO 2050 carbon emission reduction goal [12]. Thus, combining AI with excellent seamanship is crucial to developing a low-error autonomous ship system inside the overall maritime ecosystem to gain the trust of shipping companies in MASS.

The rest of this article is organized as follows: crucial related works regarding the predicted area of danger (PAD), domain-based PAD (DPAD), ship domain, and collision avoidance route planning will be introduced in Section 2. The marine collision avoidance route planning (MCARP) model, which includes the least course deviation (LCD) model, DPAD model, Davis ship domain model, and DPAD coordinate system conversion, will be explained in Section 3. The MCARP model will be verified with a six-ship encountering scenario using ArcGIS and Transas ship simulators in Section 4. Finally, the conclusion and suggestions for future work will be elaborated in Sections 5 and 6.

## 2. Literature Review

Since the 1970s, marine navigation scholars and experienced navigators have proposed different concepts for the ship domain as a practical method to depict ship collision avoidance [13]. The ship domain is a collaboration between marine traffic engineering and International Regulations for Preventing Collisions at Sea (COLREG) rules. It can be considered a fictional area around a ship that is clear of other obstacles. When an OOW must make precise decisions to avoid colliding with target ships, the ship domain can be a decent reference. It implies that a ship is in danger of collision and immediate actions must be taken if the distance at closest point of approach (DCPA) of a target ship is shorter than the region of the ship domain. The Goodwin ship domain model, as shown in Figure 1, was formulated using statistical data on marine traffic density in the Strait of Dover [14]. There are three asymmetric sectors in the Goodwin ship domain model. They refer to the required collision avoidance distance between the own ship and target ships from different bearings. Sector 1 is between 0° and 112.5° of the own ship's relative bearing, the radius of which is 0.85 nautical miles (nm). Sector 2 and Sector 3 are between 247.5° and 360° and 112.5° and 247.5°, respectively, with radii of 0.7 nm and 0.45 nm, respectively. The distances and bearing ranges in the Goodwin ship domain model are in accordance with the COLREG rules the own ship must make for the target ships on her starboard side, while ships on the port side or aft side should be given less attention.

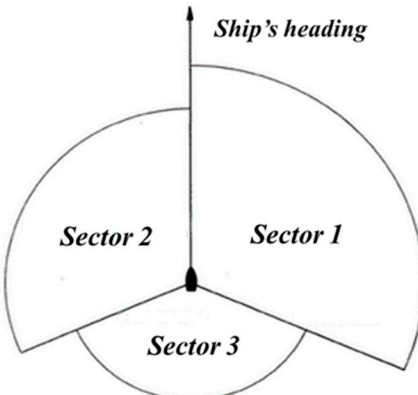

**Figure 1.** Goodwin ship domain model.

Eventually, the Goodwin ship domain model evolved into the Davis ship domain model [15], which is a smoothed ship domain as shown in Figure 2. Based on the sizes of the three sectors in the Goodwin ship domain model, the Davis ship domain model replaces the three sectors with one smoothed major sector that has the same total area and nature. To maintain the unique characteristics of the Goodwin ship domain model, the three sectors in the Davis ship domain model have the same relative bearings as the ones in the Goodwin ship domain model, and the own ship is located on the bottom-left side of the central phantom ship in the Davis ship domain model. The radius of the Davis model is 0.675 nm, and the own ship is on the relative bearing of 199° from the perspective of a phantom ship, with a distance of 0.425 nm. The lateral distance between the two ships will thus be approximately 0.14 nm. In fact, there are many more types of ship domains that have been proposed by academics. However, the Davis ship domain model is the first model that adhered to the COLREG rules, which is applicable in every navigation encountering situation once it is alterable in radius.

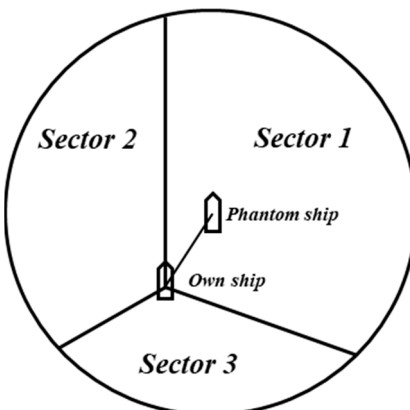

**Figure 2.** Davis ship domain model.

The process of ship collision avoidance is composed of complex decision-making and arithmetic for seafarers and marine navigation electronics. In recent years, developing the automatic collision avoidance system (ACAS) has been the focus of both the shipping industry and academia. The ACAS originated from the concept of the possible point of collision (PPC) [16], which is the intersection point of two ships' true motion lines on a radar. In the following years, an elliptical predicted area of danger (PAD) for the Sperry ACAS-I system was proposed [17], being first introduced to the Sperry Marine ARPA. The ARPA system was able to plot the PPC and PAD according to different locations, speeds, and courses of the target ship relative to the own ship by analyzing the required ship's motions. Hence, the PAD and PPC are handy tools that assist with collision avoidance decision-making. The hexagonal PAD was then introduced to the Sperry ACAS-II system, and it was based on the concept of the relative motions of a ship. The hexagonal PAD represents the predicted collision area, which possesses a shorter distance than the DCPA for both ships. The advantage of the PAD is that it makes it easier for the OOW to identify the threat that target ships pose to the own ship. Lastly, a ship-shaped PAD was proposed. The front side of the ship-shaped PAD is shaped like a half-hexagon, which indicates to the OOW that their ship is passing the head of the target ship. The rear side of a ship-shaped PAD has a half-elliptical shape. It also combined with the PPC curves to provide insight to the OOW about the speed ratio between ships. The advantage of a ship-shaped PAD is that it lets seafarers easily recognize and perform necessary collision avoidance actions immediately and effectively. However, it has never been implemented on any commercial ARPA system or tested for its performance in practice. Thus, although many PAD systems have been proposed by scholars, the hexagonal PAD was still the most widely used ARPA system in the past. The domain-based PAD (DPAD) was then invented by [18] to incorporate the Davis ship domain model into the PAD to plot the potential collision danger area more accurately. Nevertheless, since the application of the PAD on ARPA came to an end, there has been no major development or practical application of the PAD in academia or industry [19].

In recent years, there has been a growing push for academics to delve into more safety processes of collision avoidance on fully autonomous ships or regular merchant marines, such as the collision avoidance route planning scheme under the ECDIS system developed in [20]. PADs were seen as the obstruction area along a preplanned route, and the minimum bounding rectangle (MBR) and genetic algorithm (GA) were combined and utilized to search for the optimal collision avoidance route. Path planning for autonomous ships considered maneuvering ability and COLREG rules and was conducted in [21], while the fuzzy adaptive proportion-integral-derivative (PID) was operated in the course control system of the ship. A multi-ship encounter was used to verify the system. In [22], a practical collision avoidance path planning, which applied an ant colony algorithm (ACA), was constructed using GIS. The results showed that an ACA found the optimal collision avoidance route with better accuracy and efficiency, which could significantly reduce the

workload of the OOW. Another study found that the particle swarm optimization (PSO) algorithm could also be used in the collision avoidance route planning of ships: a collaboration between ship domains and PSO validated that the consistency and compatibility of different marine traffic density scenarios were acceptable [23]. Ref. [24] reviewed the close-range collision avoidance path planning algorithm in navigation and concluded that most path planning algorithms had a common disadvantage of ignoring the ever-changing nature of environmental conditions and detailed COLREG rules. On the other hand, the dynamic anti-collision A-star (DAA-star) algorithm was implemented in [25] to simulate the multi-ship encountering situations based on oval ship domains, and the results disclosed that dynamic route planning is feasible and effective considering that the navigational obstacle and collision danger areas are constantly changing. Ref. [26] used a double deep Q network (DDQN) as an automatic collision avoidance method for multi-ship encountering situations. Imazu problems verified that the algorithm can formulate effective and human-level collision avoidance strategies in a 19-ship encountering scenario.

To consider the dynamic nature of ship collision avoidance, AIS information obtained at very high frequency can be a decent source of navigation parameter input for an automatic collision route planning model that considers route planning based on collision danger areas [27]. Dynamic AIS information can be updated within 2 to 10 s according to the speed of vessels that are underway, and it can be received within 40 nm of the range [28]. Compared with ARPA information, AIS information is less impacted by weather conditions, and there are fewer target loss situations on an ECDIS monitor [29]. More voyage information such as the name and call sign of ships are presented in AIS information, and there are five times the number of target quantities that are visible [29]. Nevertheless, the navigation parameters input from ARPA can serve as a source to verify the AIS information before the system sketches the routes and DPAD automatically on the ECDIS under the condition that it is within ARPA range (24 nm) and in decent weather [30].

In conclusion, most previous studies regarding collision avoidance route planning did not have systematic ways to sketch an optimal collision avoidance route. The route planning algorithms used in previous studies are mainly based on the shortest path principle, and the ship domains or PADs used in the simulation do not consider COLREG rules. The navigation practicality and feasibility of the produced routes can thus be doubted in real-life collision avoidance situations. Thus, this study proposes the MCARP model as a theoretical concept, which combines the LCD and DPAD models from the perspective of seafarers who have real navigational experiences. The LCD model considers COLREG rules by applying the uniform course direction principle and the optimal course deviation principle, which searches for waypoints that can create the least detours and minimum safety deviations on the routes. The DPAD model is improved in its geometric characteristics, which makes the model better conform to the COLREG rule. The model and simulation results can serve as references for ECDIS developers. It can be expected that once MCARP has become the IMO mandatory automation module in the ECDIS that can be compatible with AIS information, the development speed of MASS technology will be enhanced and, in turn, the pressure placed upon the OOW or the remote controlling personnel on the shore side will be lowered by providing effective collision avoidance route references.

## 3. Marine Collision Avoidance Route Planning Model

The details of MCARP are shown in Figure 3. It consists of the decision-making and sketching processes of DPAD vertices and PPC points. The least course deviation (LCD) model is also included in MCARP as a crucial decision support tool to avoid DPADs based on the optimal route with minimum detours.

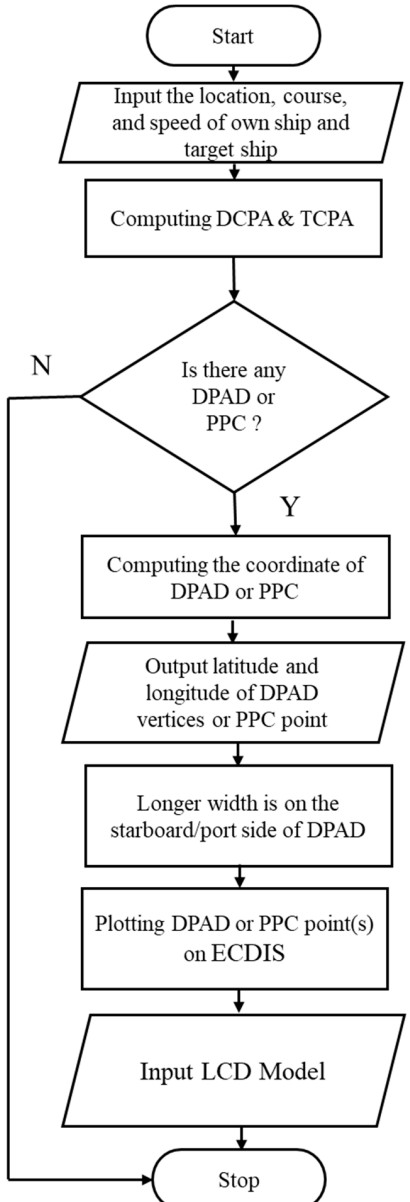

**Figure 3.** Process flow of MCARP.

### 3.1. The Least Course Deviation Model

Safety and efficiency of ship route planning are essential before sailing. When the initially planned route intersects with an unnavigable area, the navigator has to alter the waypoints of the initial route to make the altered route safe and efficient based on minimizing the detouring distance and course deviation as much as possible. However, manual alteration of waypoints on the ECDIS is sometimes complicated and even neglected or lost due to human errors, which will cause sailing to be unsafe or dangerous. Likewise, when autonomous ships sail out to sea, effective automatic route planning is also needed. Consequently, the least off course model (LOC model) was invented by [31], which is suitable for obstacle avoidance route planning for ships. This study proposes a model better suited for ship collision avoidance, the LCD model, to stimulate the automation of collision avoidance route planning. The model considers the minimum safe DCPA and the vertices of the DPADs are seen as the first/second priority waypoints. When the initial ship route goes through DPADs, the model analyzes the right/left side of the DPADs by connecting the vertices on each side with the starting point (or waypoint) automatically. Once the course deviation is calculated for each detour route, the first/second priority

collision avoidance route can be decided immediately. There are two basic principles in the LCD model. First, the optimal course deviation principle assists the system in finding the best waypoint for the next route leg. The chosen waypoint simultaneously makes the ship have the least safety course deviation and the slightest detour in avoiding obstacles. Second, the uniform course direction principle rejects the potential route leg, which crosses the obstruction unnavigable area or primary test line. The LCD model is likely to conform to COLREG rules by operating on large scales with ample time dynamically if it can be combined with an efficient route planning algorithm and navigation parameters from AIS information. It should also be noted that the LCD model can be applied to static and dynamic navigation obstacles or DPADs. The detailed process of the LCD model is shown in Figure 4.

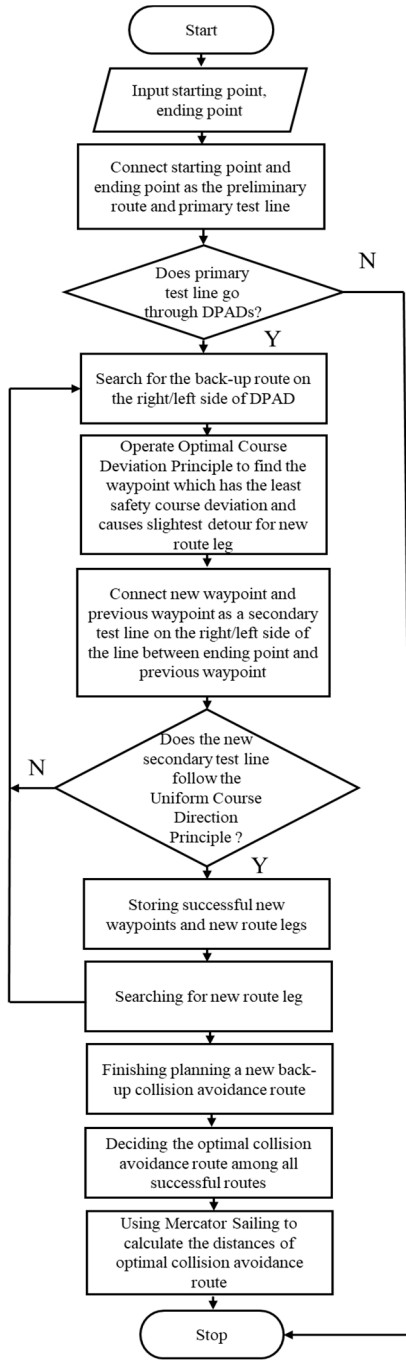

**Figure 4.** Process flow of the LCD model.

To illustrate the LCD model, a simple example of an LCD model with unnavigable areas as static obstacles will be demonstrated in the following two steps:

(1)   Step 1:

First, set point S as the starting point, and let point E be the endpoint. Next, the system creates line $\overline{SE}$ in the LCD as the primary test line. After analyzing the collision avoidance route on the starboard side in Figure 5a, the optional obstacle avoidance route 1 connecting waypoints S, A5, and E is constructed, and then the port side obstacle avoidance route is evaluated. The implementation areas of the LCD model for port-side collision avoidance routes are presented as shallow blue areas in Figure 5a–e. According to the optimal course deviation principle, Point S and Point A2 should be connected first on the port side. However, as route leg S-A2 trespasses obstruction B, the right side of area B will be detected first. Then, route leg S-B3 is connected and stored. The system thus connects point B3 and point E as the secondary test line.

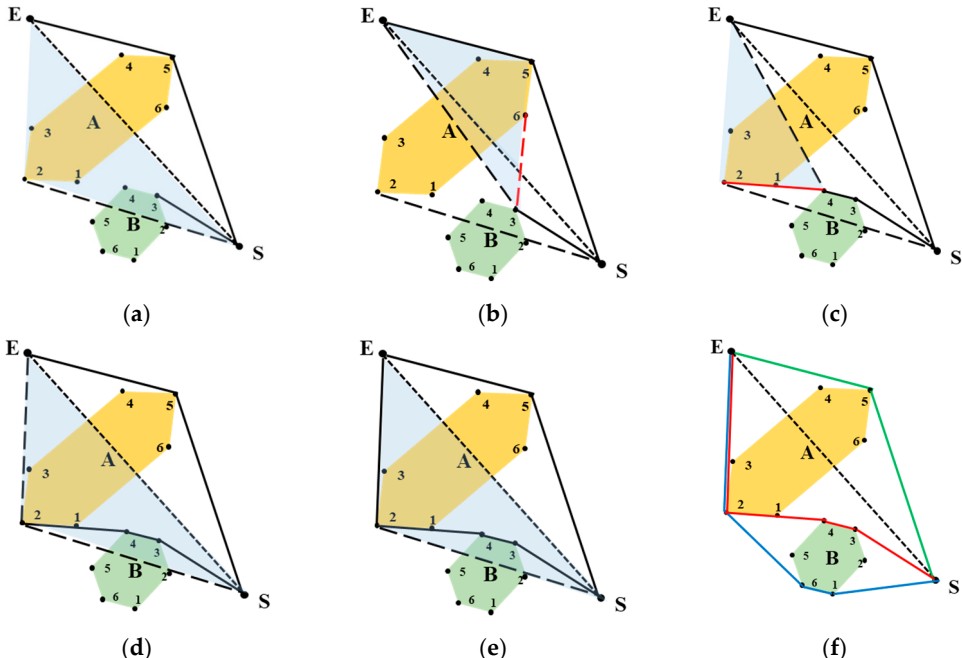

**Figure 5.** LCD model demonstration: (**a**) Route leg S-B3; (**b**) Uniform course direction principle; (**c**) Route legs B3-B4 and B4-A2; (**d**) Route leg A2-E; (**e**) Route leg A2-E; (**f**) All routes.

Moreover, since test line B3-E went through obstruction A, the system should analyze the right side of test line B3-E, which makes it B3-A6. Nonetheless, route leg B3-A6, which crosses the primary test line $\overline{SE}$, does not follow the uniform course direction principle in Figure 5b and should be rejected by the system. Afterward, the left side of test line B3-E should be evaluated. By applying the two fundamental principles in the LCD model, route leg B3-B4, route leg B4-A2, and route leg A2-E are stored, as shown in Figure 5c–e. Waypoints S, B3, B4, A2, and E formulate optimal obstacle avoidance route 2, which is the red line in Figure 5f.

(2)   Step 2:

By operating the same process based on the two fundamental principles, Waypoints S, B1, B6, A2, and E formulate optional obstacle avoidance route 3, which is the blue route in Figure 5f. The optimal obstacle avoidance route will be route 1 since the green line in Figure 5f has minimum detouring distance, and the number of waypoints is also the least. Therefore, the OOW or autonomous ship would take obstacle avoidance route 1 as a new sailing route with waypoint alterations.

### 3.2. Mercator Sailing and Parallel Sailing

The courses and distances of each rhumb line on the planning route also have to be found with a series of mathematical sailing calculations. As this study focuses on coastal navigation, Mercator sailing and parallel sailing are suitable sailing methodologies. With the acquisition of the starting point's latitude and longitude ($Ls$, $\lambda s$), the destination point's latitude and longitude ($Ld$, $\lambda d$), the difference in latitude ($l$), the difference in longitude ($DLo$), the difference in meridional parts ($m$), departure ($p$), course ($Cn$), course angle ($C$), and the Mercator sailing distance ($d$), it is possible to compute the results of Mercator ailing and parallel sailing using Equations (1)–(5).

(1)  Mercator sailing

Mercator sailing is one of the rhumb line sailing methods. Most of the prevailing nautical charts are Mercator projection charts, the meridians of which are parallel to each other. The departure between two meridians is expanded to the same length as the difference of longitude in equator as shown in Figure 6. Mercator sailing takes the discrepancies between the quasi-elliptical earth surface and plane surface into account, which applies the conformal transformation analytical function. The expansion rates of the latitude and longitude scales are sec *(L)*, keeping the Mercator charts with a correct angular relationship. In other words, Mercator sailing is the corrected version of plane sailing. It considers the concept of Meridional parts (*M*), which expands the difference in latitude between a certain latitude point and the equator on a specific meridian, but the conformal transformation makes the ship's course appropriate.

$$M = a \ln \left[ \tan 45° + \frac{L}{2} \right] - a \left( e^2 \sin L + \frac{e^4}{3} \sin^3 L + \ldots \right) \tag{1}$$

$$m = M_1 - M_2 \tag{2}$$

$$p = DLo \times \frac{l}{m} \tag{3}$$

$$d = l \cdot \sec C \tag{4}$$

where *L* is the latitude of the ship; *a* is the length of the equator's one minute of arc; and *e* is equal to the eccentricity of the earth $\approx 0.01671$.

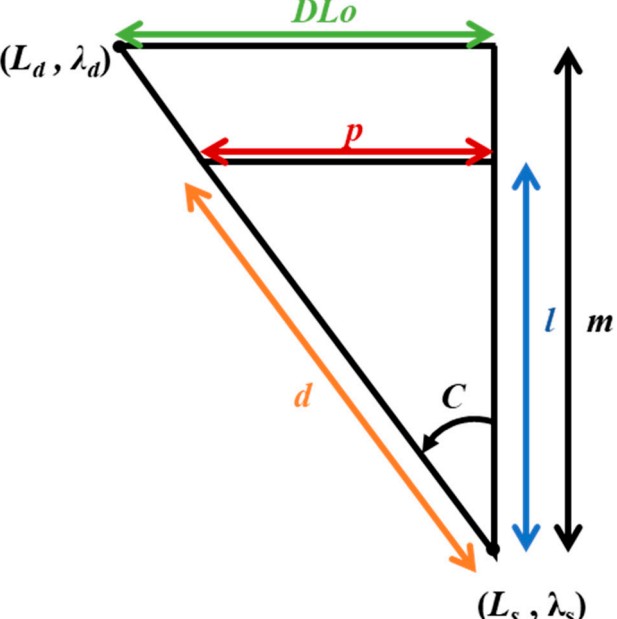

**Figure 6.** Mercator sailing.

(2)    Parallel sailing

Parallel Sailing is used when a ship is sailing due east or due west within 600 nm [32], the course of which is 090° and 270°. Because the latitude is constant while the ship sails, the difference in latitude is 0°. The sailing distance (*d*) will be equal to departure (*p*), as shown in Equation (5).

$$p = d = DLo \cdot \cos L \tag{5}$$

*3.3. Domain-Based Predicted Area of Danger*

(1)    PAD geometric model

The prerequisite to produce the PAD model on an ARPA monitor is based on the assumption that the own ship maintains its speed and each target ship keeps her course and speed in navigation practice. A PPC point is a potential point of collision between two ships. The whole PAD represents an area with a smaller DCPA than the preset DCPA. The front vertex and the rear vertex are intersection points of the own ship's course line and a target ship's course line, as in Figure 7. Therefore, when the own ship trespasses into the PAD while maintaining speed, there might be a danger of collision between the two ships.

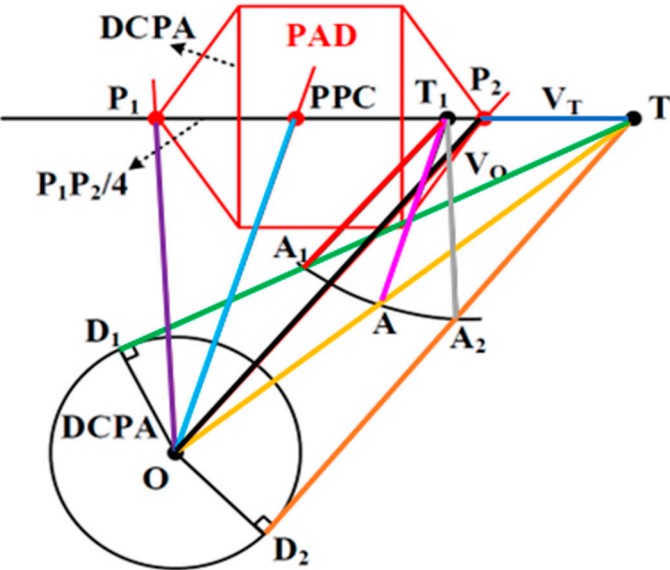

**Figure 7.** Geometry of the PAD model.

The plotting theory for the PAD is relatively simple, resembling drawing on a radar plotting sheet. For example, in Figure 7, assuming the own ship is at point O, the own ship's velocity vector $\overrightarrow{AT_1}$ (pink line) is $V_o$, and the target ship is at point T with velocity vector $V_T$, i.e., $\overrightarrow{TT_1}$(deep blue line). $\overline{OT}$ (yellow line) is the relative motion line of two ships when DCPA = 0. $\overline{TD_1}$ (green line) and $\overline{TD_2}$ (orange line) are the relative motion lines of the own ship passing through the bow and aft of the target ship with the preset DCPA. If the target ship maintains its original heading and speed, then point $T_1$ can be regarded as the center of the DCPA circle with a radius of Vo ($T_1$ is the position of the target ship after one unit of time). $\overline{TD_1}$, $\overline{OT}$, and $\overline{TD_2}$ will intersect with the unit speed vector circle of the target ship at points $A_1$, A, and $A_2$, respectively. Afterward, panning those three true motion lines $\overrightarrow{A_1T_1}$ (thick red line), $\overrightarrow{AT_1}$, and $\overrightarrow{A_2T_1}$ (grey line) to the point of the own ship O. When the ship is heading in the direction of $A_1T_1$ or $A_2T_1$, the own ship's heading line and the target ship's heading line will intersect at $P_1$ and $P_2$. On the other hand, when the ship is sailing through the course $\overrightarrow{AT_1}$, the own ship's heading line and the target ship's heading line will intersect at the PPC point.

$P_1$ and $P_2$ are two end vertices of the hexagonal PAD. When the own ship's heading passes $P_2$ or $P_1$, it shows that the own ship and the target ship will pass each other at a safe distance. However, if the own ship's course crosses into the hexagonal PAD or even goes through the PPC point, then the own ship cannot keep a safe avoidance distance when encountering the target ship. Therefore, when the navigator wants to formulate collision avoidance strategies, they should choose a course according to the new course line, such as $\overline{OP_1}$ (purple line) or $\overline{OP_2}$ (thick black line) to avoid the PAD. $\overline{OP_2}$ is the optimal new course line for the own ship since it will pass by the aft of the target ship according to the recommended collision avoidance manner of COLREG. In an exceptional situation, when the new course line, such as $\overline{OP_2}$, with a proper DCPA distance slightly crosses into the PAD of the other ship, the course line should be called the auxiliary line, and the own ship should increase the deviation angle with the line if it needs to safely avoid collision.

It can be seen from the above-mentioned PAD plotting theory that the construction of the own ship's PAD is based on the target ship information acquisition by the ARPA system, which allows the navigator to decide the safety deviation in the course easily. By obtaining accurate and real-time navigation parameters of the AIS into the ECDIS using a specific sensor, PADs can be calibrated and updated within seconds and be compared with ARPA information on the ECDIS monitor instantly and automatically. Thus, the ECDIS will play an important role in implementing the PAD in the future.

(2)　DPAD geometric model

To make the PAD more accurately displayed, [20] proposed that a PAD model should be combined with the ship domain, which makes it the domain-based predicted area of danger (DPAD), as shown in the model in Figure 8. With the acquisition of the location, speed over ground (SOG), and course over ground (COG) of the target ship from AIS information in the ECDIS system, the plotting theory for the DPAD is quite similar to the PAD. Nevertheless, in the model-building process for the PAD, the center point and the variable DCPA circle are in the own ship's location and normally have a radius of 2 nms for the DCPA. In navigation practice, due to COLREG rules and the varying maneuvering characteristics of each ship, the distances of safety collision avoidance toward target ships are considered different from different bearings. Thus, a 2 nms radius for the DCPA circle may be too large, especially when sailing in narrow and busy waters or port areas.

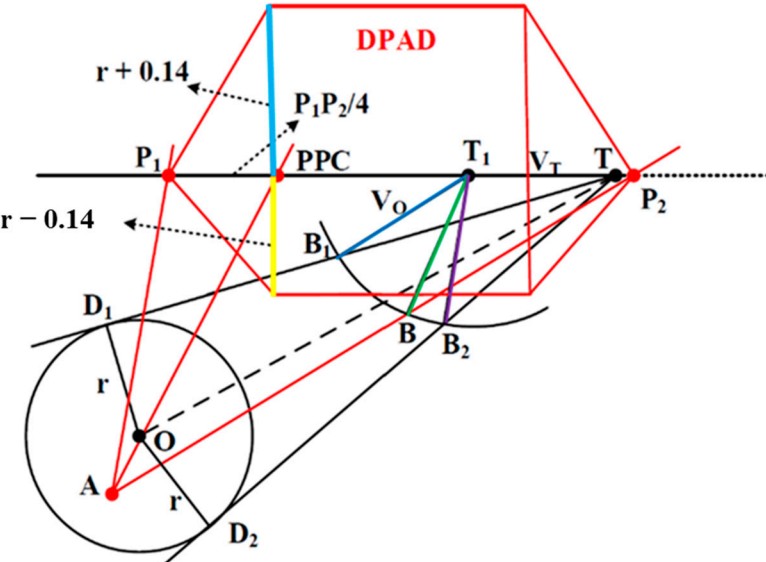

**Figure 8.** The DPAD model.

Thus, the DPAD model adopted the Davis ship domain model to retain its advantage., thus making it appropriate for ameliorating or plotting the DPAD model. The shape of

the DPAD is distinctive from the PAD. Three true motion lines $\overrightarrow{B_1T_1}$ (deep blue line), $\overrightarrow{BT_1}$ (green line), and $\overrightarrow{B_2T_1}$ (purple line) of the own ship are translated to point A (location of the own ship) and then intersect with the target ship's course line at points P1, P2, and PPC. The DPAD will be sketched according to the parameters of the DCPA (such as radius r in Figure 8).

Unlike the DPAD width nature put forward in [18], which indicates that the width closer to the own ship side is longer, this study proposes the starboard side width (shallow blue line) of DPAD as r + 0.14 nm, and port side width (yellow line) is thus r − 0.14 nm. The width nature of the DPAD follows the lateral distance between the phantom ship and the own ship in the Davis ship domain model. The starboard side width is larger because, as stated in the theory of the ship domain model, the own ship has to make way for the target on her right-hand side, which will require a bigger collision avoidance distance.

(3)　Coordinate system and the DPAD mathematical model

In the practice of marine navigation, a geographical coordinate system is utilized. The location of a ship on earth should be presented with latitude and longitude. However, for the convenience of building a DPAD mathematical model, a corresponding Cartesian coordinate system should be constructed first. In Figure 9, assuming that the geographical coordinate of the own ship and target ship are at point O $(L_o, \lambda_o)$ and point T $(L_T, \lambda_T)$, respectively, the SOG and COG of the own ship and target ship are $V_o$ (deep blue arrowed line) and $C_o$ (direction of deep blue arrowed line) and $V_t$ (purple arrowed line) and $C_t$ (direction of purple arrowed line), respectively.

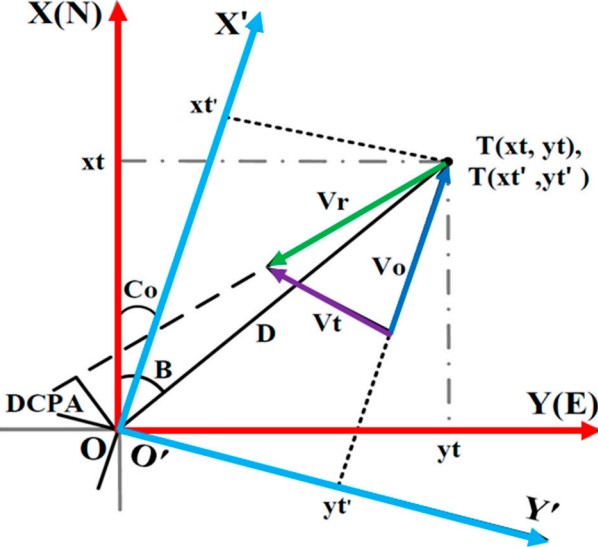

**Figure 9.** North-up and the own ship's Cartesian coordinate system.

First, converting the geographical coordinate system into a north-up Cartesian coordinate system (X − O − Y) with red arrowed lines, the X-axis points toward the due north side, and the Y-axis indicates the due east side. The location of the own ship and target ship are now at point O (0, 0) and point T $(x_t, y_t)$, respectively. The distance (D) between two ships can be obtained using the coordinates of two ships. The relative speed $V_r$ and relative course $C_r$ are acquired using the XY component ($V_{rx}$ and $V_{ry}$) of the speed vector $V_o$ and $V_t$. The DCPA and TCPA can also be derived from $B_o$, which is the relative bearing of the target ship from the perspective of the own ship.

Then, the north-up Cartesian Coordinate System axis is rotated clockwise Co degrees to the own ship's Cartesian coordinate system (X′ − O′ − Y′), indicated with light blue arrowed lines, where the heading of the own ship is the X′ positive axis direction at the moment, which is similar to the head-up navigation chart. The coordinates of the own

ship are still at the origin point, and the coordinates of the target ship can be converted by the coordinate axis rotation Equation (9) to obtain T $(x_t', y_t')$. The relative bearing is also altered to Q, but the relative speed $V_r$ (green arrowed line) and relative heading $C_r$ (direction of green arrowed line) remain unchanged.

In the north-up Cartesian coordinate system (X − O − Y), the equations for the computation of different parameters are shown in Equations (6)–(10).

$$T : \begin{cases} x_t = L_t - Lo \\ y_t = (\lambda_t - \lambda_o) \cdot \cos Lm \end{cases} \tag{6}$$

$$D = \sqrt{x_t^2 + y_t^2} \tag{7}$$

$$\vec{V_r} = \vec{V_t} - \vec{V_o} \tag{8}$$

$$\begin{cases} V_{rx} = V_t \cos C_t - V_o \cos C_o \\ V_{ry} = V_t \sin C_t - V_o \sin C_o \end{cases} \tag{9}$$

$$V_r = \sqrt{V_{rx}^2 + V_{ry}^2} \tag{10}$$

The relative heading $C_r$ between the two ships is obtained from the relative speed triangle of the two ships. The value of $C_r$ is between 0° and 180°, and the conditions for angle conversion are shown in Equation (11). The relative bearing B of the target ship, which is the bearing of the own ship clockwise from true north to the target ship, is obtained using the coordinates of the target ship on the XY axis, and its angle conversion conditions are the same as the relative course, as shown in Equation (12). In Equation (13), the relative bearing $B_o$ of the own ship from the perspective of the target ship is the relative bearing value of the own ship plus 180°. The DCPA between the target ship and the own ship can be calculated using the parameters acquired above and the TCPA, as shown in Equations (14) and (15).

$$C_r = \tan^{-1}\left(\frac{V_{rx}}{V_{ry}}\right) \tag{11}$$

$$B = \tan^{-1}\left(\frac{y_t}{x_t}\right) \tag{12}$$

$$B_o = B + 180° \tag{13}$$

If $Bo \geq 360°$, then $Bo = Bo - 360°$, and the following are used:

$$DCPA = D \cdot \sin(C_r - B_o) \tag{14}$$

$$TCPA = \frac{|D - \sin(C_r - B_o)|}{V_r} \tag{15}$$

When the coordinate system is transferred to the own ship's Cartesian coordinate system (X′ − O′ − Y′), the coordinate T of the target ship can be obtained using rotation Equation (16), and the course $C_t$ of the target ship is also converted to $C_T$ due to the rotation of the coordinate axis in Equation (17). As a result, the aspect of the target ship from the perspective of the own ship's course $C_o$ degrees is Q, and the aspect of the own ship from the perspective of the target ship's course $C_t$ degrees is Q′, as presented in Equations (18) and (19):

$$\begin{cases} x_t' = x_t \cdot \cos(C_o) + y_t \cdot \sin(C_o) \\ y_t' = -x_t \cdot \sin(C_o) + y_t \cdot \cos(C_o) \end{cases} \tag{16}$$

$$C_T = C_t - C_o \tag{17}$$

$$Q = B - C_o \tag{18}$$

$$Q' = B_o - C_t \tag{19}$$

In Figure 10, when plotting the DPAD, the own ship is located at the bottom left of the center (point A) of the Davis ship domain. There will be an eccentric own ship Cartesian coordinate system $(X' - A - Y')$, shown with purple arrowed lines, and a Phantom ship Cartesian coordinate system $(X'' - O - Y'')$, shown with orange arrowed lines. Setting the coordinate origin in the eccentric own ship Cartesian coordinate system $(X' - A - Y')$ will make the coordinate equation more concise. In the eccentric own ship Cartesian coordinate system $(X' - A - Y')$, the distance $\overline{OA}$ (shallow blue arrowed line) between the own ship and phantom ship is 0.425 nm, which can be variable due to the condition of the navigation environment. When the coordinates are converted to the phantom ship Cartesian coordinate system $(X'' - O - Y'')$, the target ship's coordinate T can be obtained with translation Equation (20), and then the distance d between the target ship and point O can be calculated using Equation (21). Coordinate T only needs to be used when calculating parameter $\omega$. In the $\Delta$OAT (the triangle surrounded by shallow blue line, shallow green line, and dark red line), since the aspect of the own ship in Figure 10 is known to be 19°, $\angle$OAT (the angle between light blue line and dark red line) equals the relative bearing of the target ship deducting 19°. Then, to obtain $\angle$a (the angle between the pink line and light red line), $\angle$b (the angle between the grey line and dark green line) and $\angle$c (the angle between the yellow line and dark red line), the ship domain radius r in $\Delta$OTD$_2$ (the triangle surrounded by the light green line, dark green line, and dotted light red line) and the distance d (shallow green line) between the target ship and the origin are used. $\angle$OTD$_2$ (the angle between the light green line and dark green line) is obtained using Equation (23).

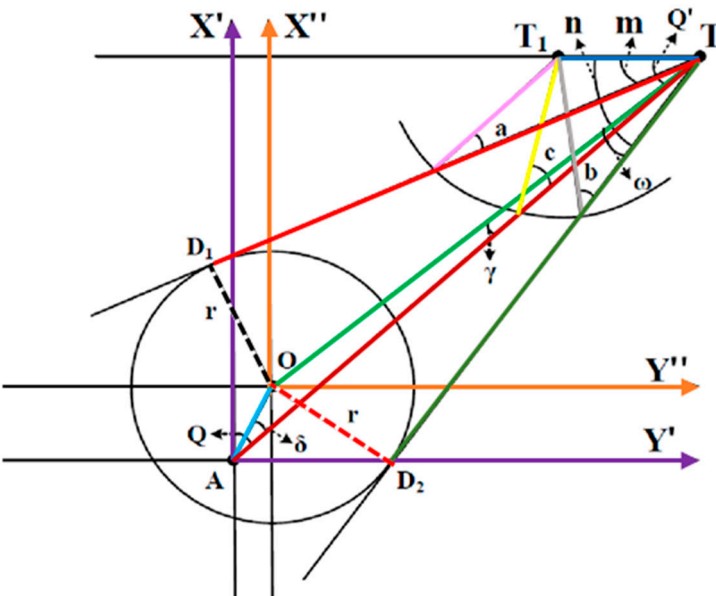

**Figure 10.** The phantom ship Cartesian coordinate system.

The aspect of phantom ship $\lambda$ (the angle between the dark blue line and light green line, which equals to $\angle T_1$TO) is acquired using Equation (24), which is the phantom ship's bearing from the fore and aft line of the target ship. $\angle$OTA (the angle between the light green line and the dark red line) is obtained using the inverse trigonometric functions of $\Delta$OAT (the triangle surrounded by the light green line, dark red line, and light blue line) in Equation (25). $\angle$a and $\angle$b are derived from $\angle T_1 TD_1$ (the angle between the dark blue line and light red line) and $\angle T_1 TD_2$ (the angle between the dark blue line and dark green line).

Finally, it is possible to calculate the two courses, $C_1$ and $C_2$, which pass through endpoints $(E_1, E_2)$ of the DPAD in Figure 11 using Equations (31) and (32), and the dangerous course $C_3$, which goes through PPC point in Equation (33). The course value ($C_1$, $C_2$, and $C_3$) at the moment is a value in the eccentric own ship Cartesian coordinate system ($X' - A - Y'$), not the actual new course value in the Geographic Coordinate System.

$$T : \begin{cases} X = x_t' + (0.425 \cdot \cos 199^\circ) \\ Y = y_t' + (0.425 \cdot \sin 199^\circ) \end{cases} \tag{20}$$

$$\overline{OT} = d = \sqrt{X^2 + Y^2} \tag{21}$$

$$\angle OAT = \delta = Q - 19^\circ \tag{22}$$

$$\angle OTD_2 = \omega = \sin^{-1}\left(\frac{r}{d}\right) \tag{23}$$

$$\angle T_1 TO = \lambda = |Q'| - \gamma \tag{24}$$

$$\angle OTA = \gamma = \sin^{-1}\left(\frac{0.425 \cdot \sin \delta}{d}\right) \tag{25}$$

$$\angle T_1 TD_1 = m = \lambda - \omega \tag{26}$$

$$\angle T_1 TD_2 = n = \lambda + \omega \tag{27}$$

$$\angle a = \sin^{-1}\left(v_t \cdot \frac{\sin(m)}{V_o}\right) \tag{28}$$

$$\angle b = \sin^{-1}\left(v_t \cdot \frac{\sin(n)}{V_o}\right) \tag{29}$$

$$\angle c = \sin^{-1}\left(v_t \cdot \frac{\sin(Q')}{V_o}\right) \tag{30}$$

$$C_1 = Q + \omega + \gamma - a \tag{31}$$

$$C_2 = Q - \omega + \gamma - b \tag{32}$$

$$C_3 = Q - c \tag{33}$$

When the target ship is located on the port side of the own ship, the computation method is the same. $Q'$ and $Q$ are still positive, with ranges of $0^\circ \leq Q \leq 180^\circ$ and $0^\circ \leq Q' \leq 180^\circ$, respectively. $\angle OAT$, $C_1$, $C_2$, and $C_3$ are calculated using Equations (34)–(37):

$$\angle OAT = \delta = Q + 19^\circ \tag{34}$$

$$C_1 = 360^\circ - Q - \omega - \gamma + a \tag{35}$$

$$C_2 = 360^\circ - Q + \omega - \gamma + b \tag{36}$$

$$C_3 = 360^\circ - Q + c \tag{37}$$

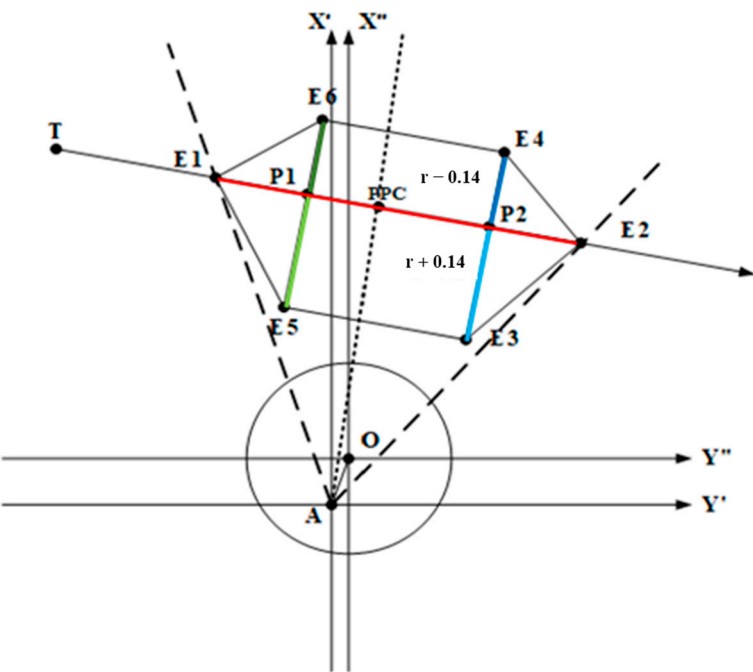

**Figure 11.** The DPAD geometrical model (Own ship is on the starboard side of the target ship).

After obtaining the safe course of the own ship, which goes through the two endpoints of the DPAD, the speed ratio k of the two ships (k = $V_o/V_t$), λ, and Q can be used to determine the quantities of the DPAD and PPC. There are 14 types of DPAD shapes according to the parameters mentioned above. Then, the vertex $E_1$, $E_2$, and PPC coordinates of the DPAD can be obtained by searching the intersection point of the own ship course line and target ship course line. Because of the geometrical nature of the DPAD, which includes $\overline{E_3E_4}$ (the combination of the dark and light blue lines) ⊥ $\overline{E_1E_2}$(red line), $\overline{E_5E_6}$ (the combination of the dark and light green lines) ⊥ $\overline{E_1E_2}$, and the eccentricity of the own ship in the Davis ship domain, $\overline{P_1E_5}$ (light green line) and $\overline{P_2E_3}$ (light blue line) on the starboard side of the target ship have a distance of r + 0.14 nm from the edge of the hexagon, and their value is 0.81 nm. On the other hand, $\overline{P_1E_6}$ (dark green line) and $\overline{P_2E_4}$ (dark blue line) on the port side of the target ship have a distance of r − 0.14 nm from the edge of the hexagon, the value of which is 0.53 nm. Therefore, the coordinates of vertices $E_3$~$E_6$ and PPC point of the hexagonal DPAD can be acquired using Equations (38)–(46).

$$PPC : \begin{cases} X_c = \frac{y_t' - x_t' \times \tan C_T}{\tan C_3 - \tan C_T} \\ Y_c = X_c \times \tan C_3 \end{cases} \tag{38}$$

$$E_1 : \begin{cases} X_1 = \frac{y_t' - x_t' \cdot \tan C_T}{\tan C_1 - \tan C_T} \\ Y_1 = X_1 \cdot \tan C_1 \end{cases} \tag{39}$$

$$E_2 : \begin{cases} X_2 = \frac{y_t' - x_t' \cdot \tan C_T}{\tan C_2 - \tan C_T} \\ Y_2 = X_2 \cdot \tan C_2 \end{cases} \tag{40}$$

$$P_1 : \begin{cases} X_{p1} = X_1 \mp \frac{X_1 - X_2}{4} \\ Y_{p1} = Y_1 \mp \frac{Y_1 - Y_2}{4} \end{cases} \tag{41}$$

$$P_2 : \begin{cases} X_{p2} = X_2 \mp \frac{X_1 - X_2}{4} \\ Y_{p2} = Y_2 \mp \frac{Y_1 - Y_2}{4} \end{cases} \tag{42}$$

$$E_3 : \begin{cases} X_3 = X_{p2} + 0.81 \cdot \cos(C_T + 90°) \\ Y_3 = Y_{p2} + 0.81 \cdot \sin(C_T + 90°) \end{cases} \tag{43}$$

$$E_4 : \begin{cases} X_4 = X_{p2} + 0.53 \cdot \cos(C_T - 90^{\circ}) \\ Y_4 = Y_{p2} + 0.53 \cdot \sin(C_T - 90^{\circ}) \end{cases} \tag{44}$$

$$E_5 : \begin{cases} X_5 = X_{p1} + 0.81 \cdot \cos(C_T + 90^{\circ}) \\ Y_5 = Y_{p1} + 0.81 \cdot \sin(C_T + 90^{\circ}) \end{cases} \tag{45}$$

$$E_6 : \begin{cases} X_6 = X_{p1} + 0.53 \cdot \cos(C_T - 90^{\circ}) \\ Y_6 = Y_{p1} + 0.53 \cdot \sin(C_T - 90^{\circ}) \end{cases} \tag{46}$$

When the CT is 90° or 270°, the value of tan CT is $\infty$ or $-\infty$. Therefore, Xc, $X_1$, and $X_2$ will not be valid values. However, when CT = 90° or 270°, Xc = $X_1$ = $X_2$ = Xt′ in the eccentric own ship Cartesian coordinate system (X′ − A − Y′). Therefore, this method can be used to quickly discover the DPAD coordinate points in this particular situation.

As the coordinates of the DPAD on the eccentric own ship Cartesian coordinate system (X′ − A − Y′) and the PPC are determined to display on the chart, it is necessary to convert the coordinates of each point back to the longitude and latitude coordinate system in order—taking PPC point (Xc, Yc) as an example—using the rotation formula in Equation (47) and the translation formula in Equation (48) to convert the coordinates of each point. The coordinate in the north-up Cartesian coordinate system is (Xc′, Yc′), and finally, the coordinate in the geographic coordinate system is (Lc, λc).

$$\begin{cases} X_c' = X_c \cdot \cos(360^{\circ} - C_o) + Y_c \cdot \sin(360^{\circ} - C_o) \\ Y_c' = -X_c \cdot \sin(360^{\circ} - C_o) + Y_c \cdot \cos(360^{\circ} - C_o) \end{cases} \tag{47}$$

$$\begin{cases} L_c = X_c' + L_o \\ \lambda_c = \left( \dfrac{Y_c'}{\cos(L_m)} \right) + \lambda_o \end{cases} \tag{48}$$

## 4. Empirical Analysis: Six-Ship Encountering Scenario

As presented in Figure 12, the scenario is to verify the LCD and DPAD models in a six-ship encountering situation, which is more complicated than the usual two-ship encountering situation. The collision avoidance situations include crossing conditions, fully head-on encountering, and a stationary ship encountering. T3, T4, and T5 are crossing ships, T1 is a fully head-on vessel for the own ship, and T2 is a fixed target ship with a heading of 140°. The six ships are all set to be stand-on 400 m container ships, the details of which are provided in Figure 13 and Table 2. The DCPA between the five target ships is 1.2 nm. No risk of collision is considered between the target ships. After using MCARP to analyze the optimal collision avoidance route in ArcGIS to avoid the DPADs, the results of the collision avoidance route analysis were tested using the Transas ship simulator at NTOU [21], as shown in Figure 14, to verify the applicability of MCARP in a real navigation environment. This demonstration only encompasses the static DPADs and collision avoidance routes without updating navigation parameters from AIS information.

**Table 2.** Information for each target ship.

| Ships | Latitude | Longitude | COG | SOG |
|-------|----------|-----------|-----|-----|
| T1 | 25°06.20′ N | 170°08.10′ E | 230° | 20 kts |
| T2 | 25°04.50′ N | 170°10.50′ E | 140° | 0 kts |
| T3 | 24°59.00′ N | 170°10.00′ E | 320° | 24 kts |
| T4 | 25°10.00′ N | 170°06.00′ E | 170° | 18 kts |
| T5 | 25°07.00′ N | 170°12.00′ E | 260° | 15 kts |

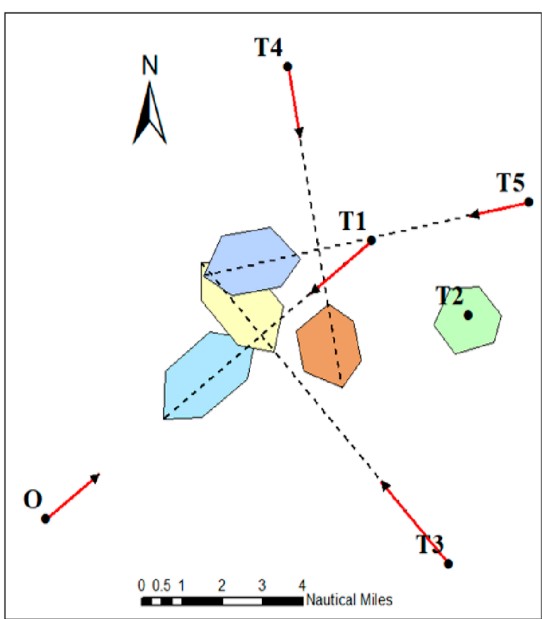

**Figure 12.** The DPAD for each target ship in ArcGIS.

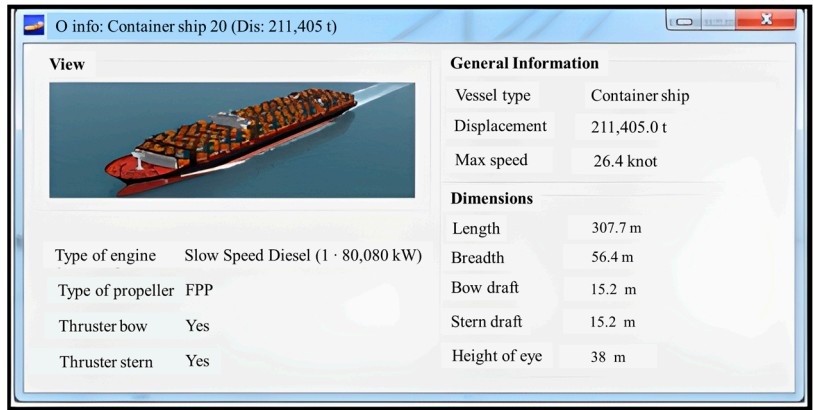

**Figure 13.** Navigation parameters for all 6 ships.

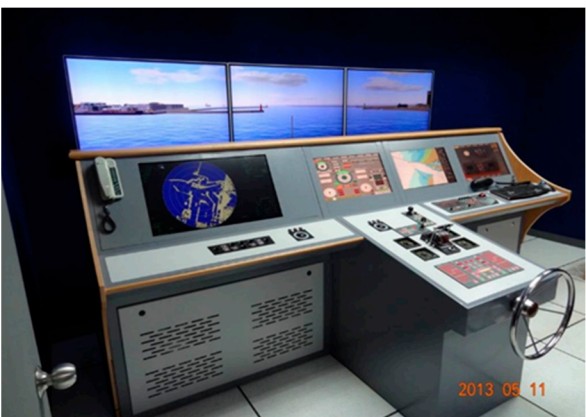

**Figure 14.** Transas ship simulator at NTOU [33].

Assuming that the starting point O is at (25° N, 170° E), and the destination point O′ is at (25°09.64′ N, 170°12.61′ E), the own ship (O) will proceed on $\overline{OO'}$ at a speed of 20 knots with a course of 50°. The DCPA was set to be 0.675 nm, which can be variable according to the preference of an OOW. After inputting the navigation parameters of six

ships into MCARP, the coordinates of each DPAD will be produced. Then, the DPADs can be automatically sketched in ArcGIS one at a time. From the distribution diagram of DPAD in Figure 12, it can be ascertained that the own ship will need to take collision avoidance action against the fully head-on ship T1 first. Afterward, if the own ship opts to turn to the starboard side, she must pay attention to the incoming ship T4 on the port bow of her and the stationary target ship T2 dead ahead of her. On the contrary, if the own ship chooses to turn to her port side, it may cause a close-quarters situation between the target ships T1, T3, and itself. The own ship will also keep a sharp lookout on target ship T5 on her starboard bow.

After formulating the DPADs in ArcGIS, MCARP will initiate the route planning stage. Once MCARP detects that the original route $\overline{OO'}$ crosses through DPADs, the LCD model is immediately activated and automatically search for the optimal collision avoidance route. The collision avoidance search result is presented on an ArcGIS map, as shown in Figure 15. The waypoint location, course angle (C), course over ground (COG), route leg distances (Dist.), and total route distance (T_Dist.) are presented in Table 3.

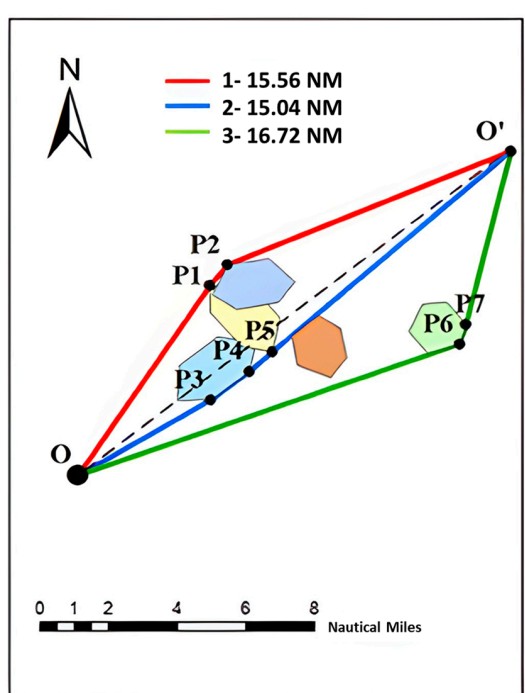

**Figure 15.** Collision avoidance routes in ArcGIS.

**Table 3.** Waypoint information.

| Waypoint Name | Route | Latitude | Longitude | C | COG | Dist. | T_Dist. |
|---------------|-------|----------|-----------|---|-----|-------|---------|
| O | 1 | 25°00.00′ N | 170°00.00′ E | N32.4° E | 32.4° | 0 | 0 |
| P1 | 1 | 25°05.60′ N | 170°03.90′ E | N34.8° E | 34.8° | 6.63 nm | 6.63 nm |
| P2 | 1 | 25°06.30′ N | 170°04.40′ E | N65.9° E | 65.9° | 0.85 nm | 7.48 nm |
| O′ | 1 | 25°09.64′ N | 170°12.61′ E | N/A | N/A | 8.07 nm | 15.56 nm |
| O | 2 | 25°00.00′ N | 170°00.00′ E | N57.9° E | 57.9° | 0 | 0 |
| P3 | 2 | 25°02.20′ N | 170°03.90′ E | N49.1° E | 49.1° | 4.13 nm | 4.13 nm |
| P4 | 2 | 25°03.10′ N | 170°05.00′ E | N43.1° E | 43.1° | 1.38 nm | 5.51 nm |
| P5 | 2 | 25°03.70′ N | 170°05.60′ E | N47.3° E | 47.3° | 0.82 nm | 6.33 nm |
| O′ | 2 | 25°09.64′ N | 170°12.61′ E | N/A | N/A | 8.71 nm | 15.04 nm |
| O | 3 | 25°00.00′ N | 170°00.00′ E | N68.9° E | 68.9° | 0 | 0 |
| P6 | 3 | 25°03.90′ N | 170°11.10′ E | N16.9° E | 16.9° | 10.84 nm | 10.84 nm |
| P7 | 3 | 25°04.50′ N | 170°11.30′ E | N13.9° E | 13.9° | 0.63 nm | 11.47 nm |
| O′ | 3 | 25°09.64′ N | 170°12.61′ E | N/A | N/A | 5.25 nm | 16.72 nm |

Subsequently, it is feasible to input the latitude and longitude of each waypoint and navigation parameters of each ship to simulate a real navigation environment. The navigation environment was designed to have no wind and no current. The simulation in the following was first operated with Transas simulation software, and it was as later verified by author 1 and author 2, who manually steered on the Transas ship simulator in Figure 16. The encountering DCPAs were measured using the variable range marker (VRM) in the ECDIS system.

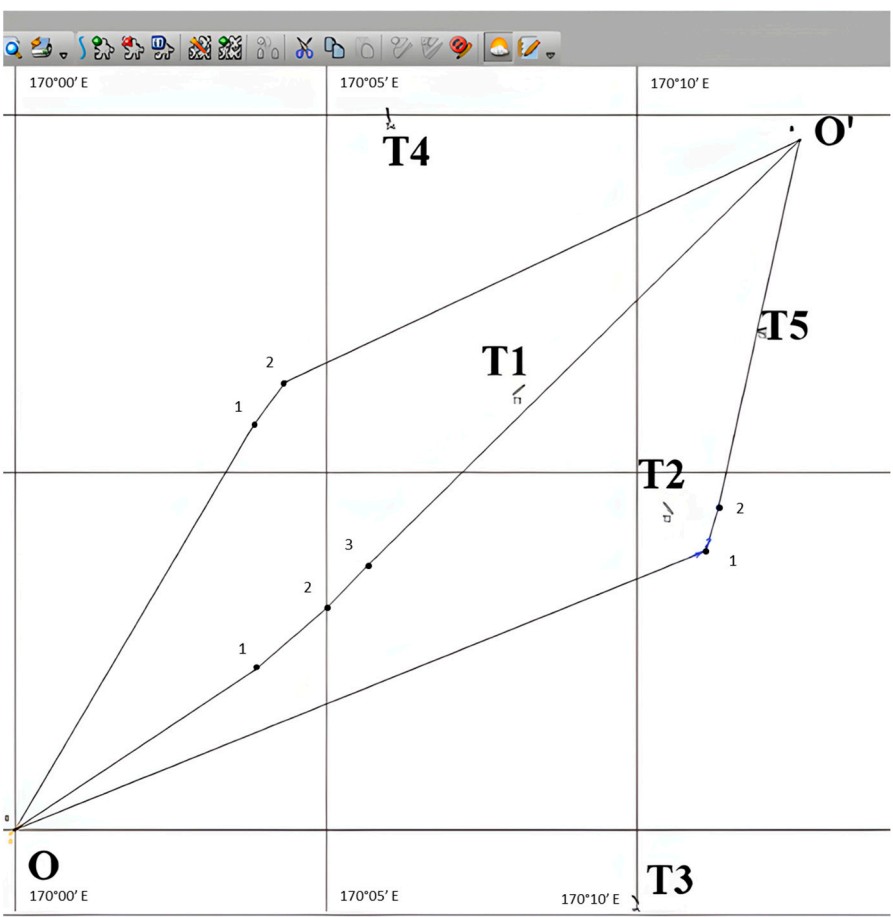

**Figure 16.** Collision avoidance routes on the Transas ECDIS System.

On route 1, the own ship passed the bow of T3 with a DCPA = 0.73 nm on waypoint P1, which is shown in Figure 17. When on waypoint P2, the own ship passed the aft of T5 with a DCPA = 0.67 nm, as presented in Figure 18. Soon after, the own ship was not at potential risk of collision. All the encountering situations in route 1 were crossing situations.

On route 2, own ship and T1 were first in a "port to port" situation with a DCPA of 0.53 nm on waypoint P3, shown in Figure 19. Then, own ship had a crossing situation facing the bow of T3 with a DCPA = 0.987 nm on waypoint P4, which is shown in Figure 20. When it was on waypoint P5, the own ship passed the aft of T4 with a DCPA = 0.987 nm, as presented in Figure 21. Finally, the own ship becomes a ship with no collision avoidance action needed.

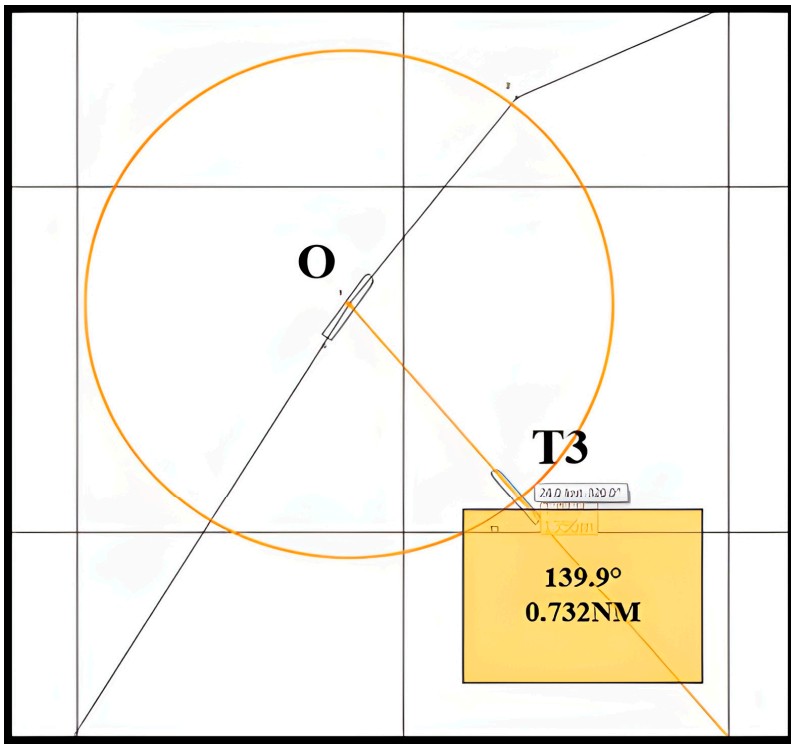

**Figure 17.** Encountering situation at waypoint P1.

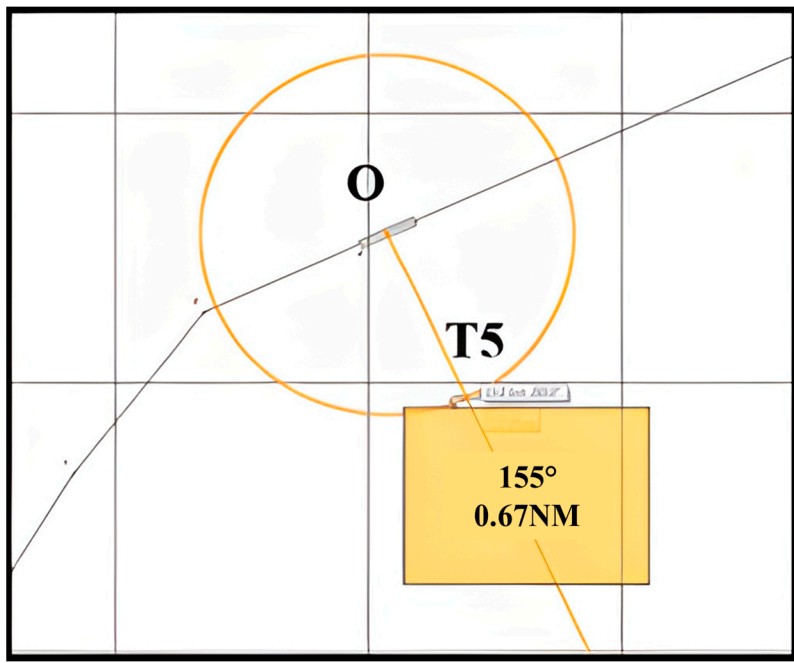

**Figure 18.** Encountering situation at waypoint P2.

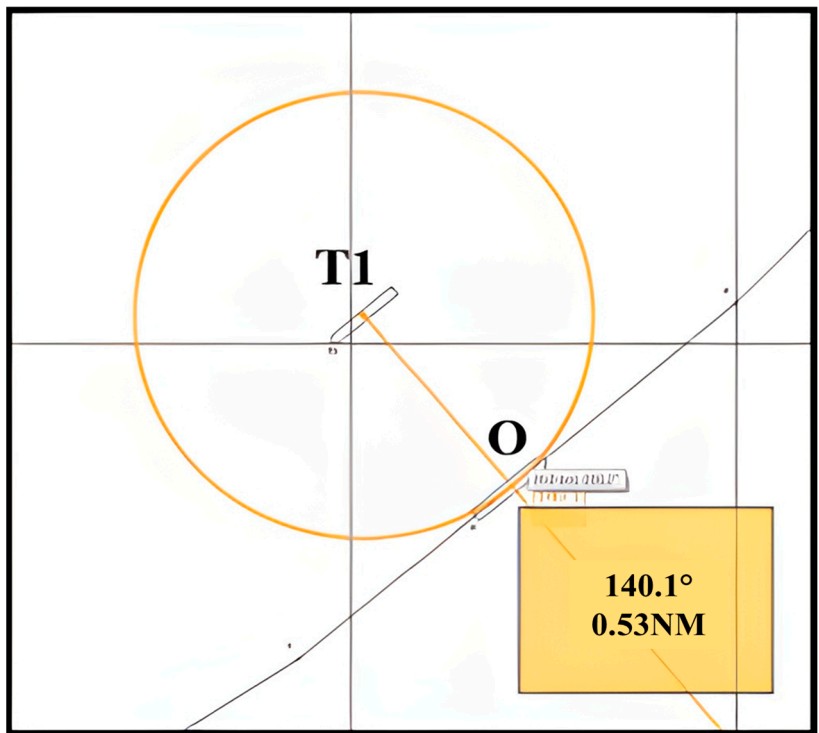

**Figure 19.** Encountering situation at waypoint P3.

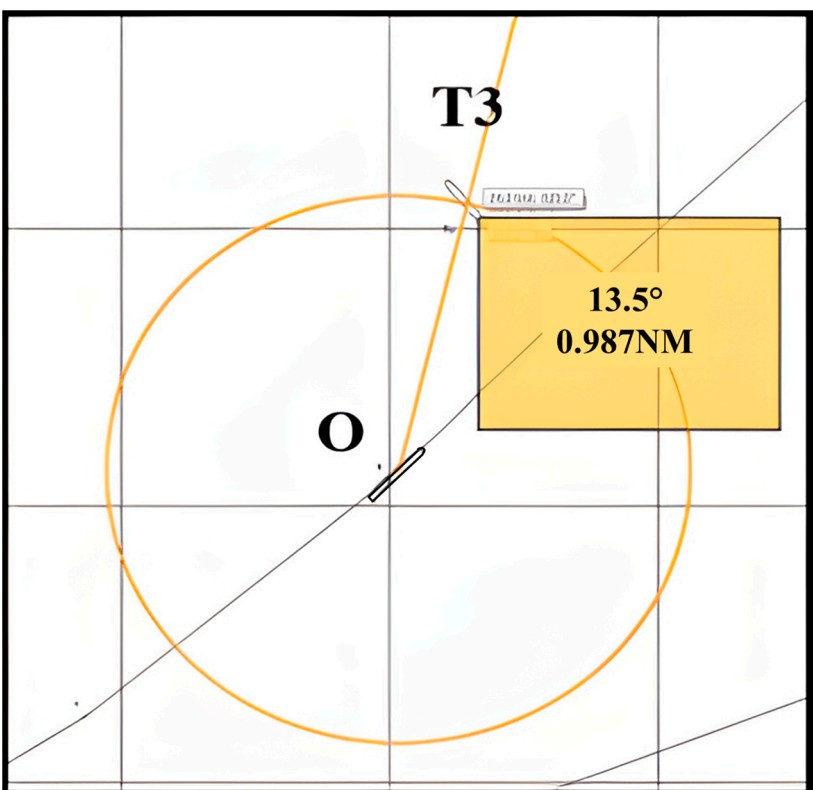

**Figure 20.** Encountering situation at waypoint P4.

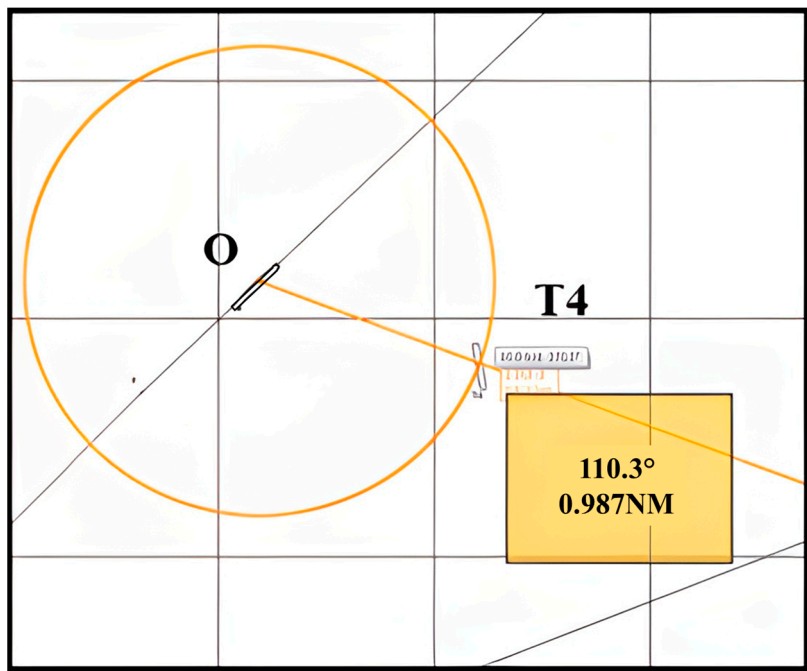

**Figure 21.** Encountering situation at waypoint P5.

On route 3, own ship passed the bow of T4 with a DCPA = 0.705 nm on waypoint P6, which is shown in Figure 22. When on waypoint P7, own ship passed the aft of stationary T2 with a DCPA = 0.774 nm, as presented in Figure 23. Subsequently, own ship was relieved of danger from the collision.

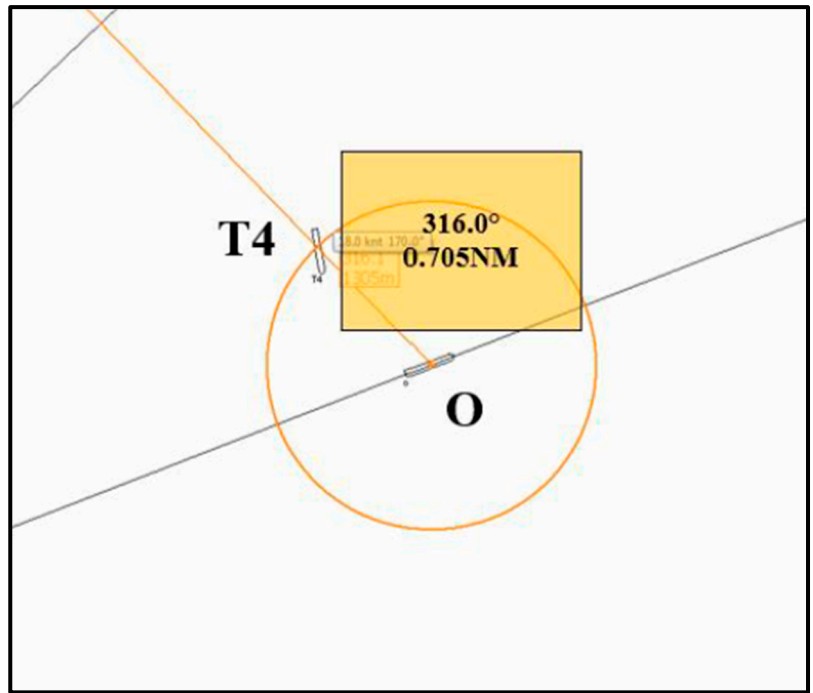

**Figure 22.** Encountering situation at waypoint P6.

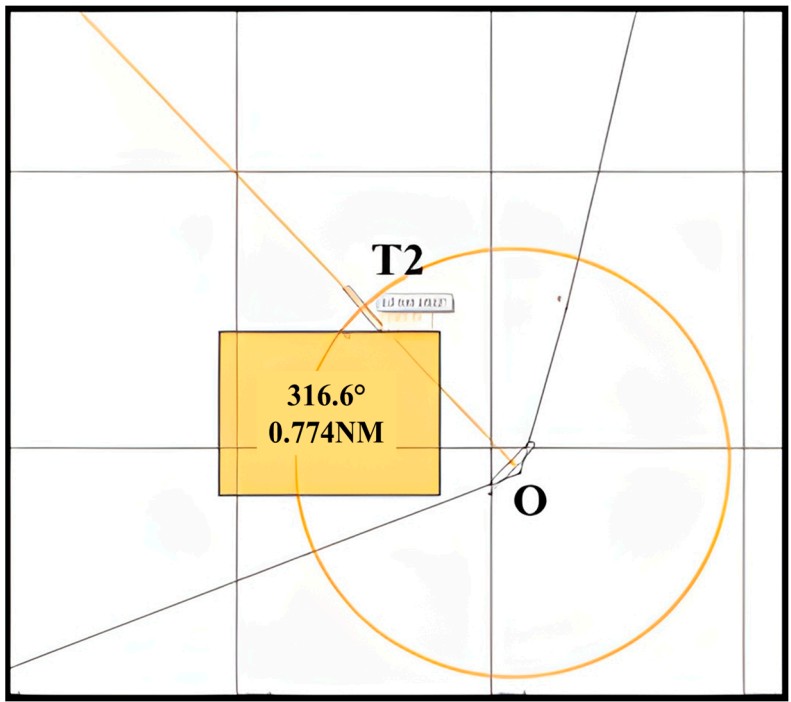

**Figure 23.** Encountering situation at waypoint P7.

To summarize the simulation, route 2 is the optimal collision avoidance route since it is the shortest route and has the fewest waypoints and a smaller rudder turning angle. Regarding the collision avoidance distance between ships, the ones in route 2 were the longest at 0.987 nm. In addition, all collision avoidance distances from waypoint 1 to waypoint 7 were larger than the safe collision avoidance distance (0.53 nm to 0.675 nm), which verified that the combination of the MCARP model and DPAD model is practically effective. It can be also expected that collision avoidance will be equivalent to human-level based on the collision avoidance performance in the simulation. Once the MCARP model becomes a mandatory module on the ECDIS, an OOW can remotely control the MASS or the MASS can automatically plan the collision avoidance route and take actions on its own.

## 5. Conclusions

This study proposes the MCARP model for the practice of marine navigation. MCARP includes the LCD model and DPAD model. Unlike the fact that most of the route planning algorithms in previous studies mainly used the shortest path principle, the LCD model can be systematically operated under the condition that the optimal collision avoidance route avoids DPADs with a uniform course direction and optimal course deviations using Mercator sailing. As a result, the optimized collision avoidance route contains fewer waypoints, an acceptable deviation in course, and minor detours. In addition, the DPAD model combines the variable Davis ship domain model and the original PAD model, implying the COLREG rules and the navigation environments are considered. This study also collaborated the DPAD model with the LCD model and performed them in a six-ship encountering scenario in ArcGIS and a Transas ship simulator. The simulation results show that the MCARP model manages to find the optimal collision avoidance route efficiently. The DCPAs on each waypoint between the own ship and the target ship are also larger than the preset collision avoidance distances (0.53 nm to 0.675 nm), proving that MCARP is applicable in ECDIS. Once the MCARP module is set up to be mandatory in the ECDIS, it will reduce the workload of the OOW, thus becoming a valuable tool for a MASS to work in autopilot mode.

## 6. Suggestions for Future Works

The LCD model is designed to analyze ship routes avoiding DPADs by choosing waypoints on the routes. However, there is much more to consider other than waypoint data when ships are sailing on the ocean; external meteorological factors, such as climates and hydrology near the preplanned shipping route are also taken into account. Therefore, to make MCARP more realistic, this study suggests that future models should be included with numerical equations for winds, currents, tides, or other weather conditions, which will fit into the ever-changing navigation conditions.

This study builds an MCARP model based on hexagonal DPADs. However, there are many different shapes of DPADs, e.g., elliptical, ship-shaped, or triangular DPADs. Therefore, this study recommends that future studies verify the performance of different DPADs with the combination of PPC curves when constructing an innovative ship collision avoidance model. Furthermore, the different maneuvering characteristics or the size of the ship domain should also be included due to the varying lengths of merchant ships. The MCARP model proposed in this study is more like a concept that can be implemented in the ECDIS with verified AIS information. Future studies can consider strategically collaborating with ECDIS developer to combine some of the renowned route planning algorithms (e.g., ant colony algorithm, particle swarm algorithm, and A-star algorithm) with the principles applied in the LCD model to demonstrate that MCARP can dynamically operate at large scales and in ample time.

**Author Contributions:** Conceptualization, C.-W.L., C.-K.H. and Y.-L.C.; methodology, C.-W.L., C.-K.H. and Y.-L.C.; software, C.-W.L., C.-K.H. and Y.-L.C.; validation, C.-W.L., C.-K.H. and C.-M.L.; formal analysis, C.-W.L., C.-M.L. and F.-S.Y.; investigation, C.-W.L., C.-M.L. and F.-S.Y.; data curation, C.-W.L., C.-M.L. and F.-S.Y.; writing—original draft preparation, C.-W.L. and C.-K.H.; writing—review and editing, C.-W.L., C.-K.H. and C.-M.L.; visualization, C.-W.L. and C.-K.H.; supervision, C.-K.H. and C.-M.L.; funding acquisition, C.-M.L. All authors have read and agreed to the published version of the manuscript.

**Funding:** This research was partially sponsored by the National Science and Technology Council, Taiwan, R.O.C., under contract MOST-112-2622-8-005-001-TE1.

**Institutional Review Board Statement:** Not applicable.

**Informed Consent Statement:** Not applicable.

**Data Availability Statement:** The data presented in this study are available on request from the corresponding author. The data are not publicly available due to confidentiality.

**Acknowledgments:** The authors would like to thank Brandon Lu, a native English speaker from New South Wales University, for helping to proofread and edit this article.

**Conflicts of Interest:** The authors declare no conflict of interest.

## Abbreviations

| | |
|---|---|
| ARPA | automatic radar plotting aids |
| ECDIS | Electronic Chart Display and Information System |
| CPA | closest point of approach |
| TCPA | time to closest point of approach |
| DCPA | distance at closest point of approach |
| ENC | electronic navigation chart |
| AIS | automatic identification system |
| OOW | officer on watch |
| IMO | International Maritime Organization |
| PPC | possible point of collision |
| MBR | minimum bounding rectangle |
| MCARP | marine collision avoidance route planning |
| MASS | Maritime Autonomous Surface Ships |
| PAD | predicted area of danger |



| | |
|---|---|
| DPAD | domain-based predicted area of danger |
| COLREG | International Regulations for Preventing Collisions at Sea |
| PID | proportion-integral-derivative |
| ACAS | automatic collision avoidance system |
| ACA | ant colony algorithm |
| PSO | particle swarm optimization |
| LCD | least course deviation |
| LOC | least off course |
| GA | genetic algorithm |
| COG and SOG | course over ground and speed over ground, respectively |

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
