# Peer review of "Marine Collision Avoidance Route Planning Model for MASS Based on Domain-Based Predicted Area of Danger"

_jmse, doi:10.3390/jmse11091724_

Round 1

Reviewer 1 Report

1. One of my major concern is that the authors failed to demonstrate the idea about how you could applied the proposed method on the decision-making and control module on MASS? The proposed method is classic ARPA-based collision avoidance method for conventional ship steered by ship officers. how can this be applied on MASS as argued in the title of the paper?

2. The authors utilized the least course deviation model in the route planning model. Will this be contradictory to the COLREGs rules that suggest the avoidance manoeuvre should be made in large scale and in ample time?

Author Response

  1. One of my major concerns is that the authors failed to demonstrate the idea about how you could apply the proposed method on the decision-making and control module on MASS? The proposed method is classic ARPA-based collision avoidance method for conventional ship steered by ship officers. how can this be applied on MASS as argued in the title of the paper?

Authors’ Response: Thank you very much for the comment and support. This study proposed the concept of MCARP in accordance with the LCD model and DPAD. The whole computational process should be implemented on ECDIS as a mandatory module through the capturing of AIS information to renew the collision avoidance routes and the positions of DPADs for OOW or the remote controlling personnel on the shore side. The navigation parameters input from ARPA can serve as a source to verify the AIS information before the system sketch the routes and DPAD automatically on ECDIS under the condition that it is within ARPA range and in decent weather. The context has been added between line 183 to 194.

  1. The authors utilized the least course deviation model in the route planning model. Will this be contradictory to the COLREGs rules that suggest the avoidance maneuver should be made in large scale and in ample time?

Authors’ Response: Thank you very much for the comment. LCD model will not be contradictory to COLREG rules since the ECDIS can capture the navigation parameters of target ships through AIS information in 40 nautical miles of range, meaning that the collision avoidance maneuver can be done in advance with enough time. The AIS information can be renewed in every 2 to 10 seconds if the ships are underway, implying that LCD model can be operated in large scale. With this scheme, LCD model will become a decision support tool for navigators or remote controlling personnel on the shore side to automatically formulate collision avoidance strategies if it can be integrated into the ECDIS system as a mandatory automation module especially for MASS. This study used ArcGIS as the collision avoidance route generation platform, and the collision avoidance results on ECDIS showed that once the routes can be produced quickly and accurately, the avoidance maneuver can be effective in large scale and ample time. The context has been added between line 241 to 243.

Reviewer 2 Report

Marine Collision Avoidance Route Planning Model for MASS
based on Domain-based Predicted Area of Danger

De article has a clear objective and a clear structure. The problem is recognised by both the scientific world and the nautical world.

The approach as such not new. Automated detection of conflict area's is known for at least since 2011, and shortest path methods have been studied for decades, but the combination is not studied in depth yet. This is a next step in that area.

For the technique of conflict detection, an approach has been chosen not widely accepted or applied in the maritime world (PAD). This may conflict with ships that use other techniques.

The manuscript has a few shortcoming. In the introduction, around line 70, the authors claim that automated techniques are superior to human decision making. I am afraid the authors do not have data that supports this claim. Navigators in general are extremely good in resolving conflicts, as shown in the low accident rate with respect to collisions. Automated systems have not been put to the test yet.

Figure 5 (page 7) shows route optimisation. The result can be true for static objects, but will be incorrect for moving objects. From the text it is not clear what kind of objects they are.

Figure 7 (page 9) is not clear. Distances, vectors and lines cannot be distinguished. All black lines may be correct, but the colored lines raise several questions. The text does not resolve this. The same for figure 8. As a result, I cannot verify if the conclusions are correct.

The calculations on page 11 to 15 are very different from the approach I am used to. Because it was hard to follow, I did not verify if all is correct.

Page 16 starts describing a simulator experiment. In table 4, the position of T3 is incorrect. The navigator of O maintains a minimum CPA of .675NM. The ships T4 and T5 pass at a smaller distance. Manoeuvring of the T-ships can be expected, possibly changing the entire outcome. Compare this also with the claim about superior automation vs. humans.

The manuscript has many small language mistakes. None of them makes it impossible to understand the meaning, but they must be corrected. This can be realised easily by a native speaker.

Author Response

  1. For the technique of conflict detection, an approach has been chosen not widely accepted or applied in the maritime world (PAD). This may conflict with ships that use other techniques.

Authors’ Response: Thank you very much for the valuable comment and support. This study proposed the concept of MCARP based on the DPAD and LCD models. The ideal implementation scenario is that MCARP model can be programmed and input into ECDIS as a mandatory module regulated by IMO. With this condition, every MASS will possess the same collision avoidance route planning decision support tool whether for navigators or remote controlling personnel on the shore side. The context has been added between line 208 to 212.

  1. The manuscript has a few shortcomings. In the introduction, around line 70, the authors claim that automated techniques are superior to human decision making. I am afraid the authors do not have data that supports this claim. Navigators in general are extremely good in resolving conflicts, as shown in the low accident rate with respect to collisions. Automated systems have not been put to the test yet.

Authors’ Response: Thank you very much for the valuable comment. We all agreed with the point of view from the reviewer. We have also found that there have sea trial results of fully autonomous ship which have been published in the journal paper as specified in the source 11. Moreover, to make the claim more reasonable, the context have been revised between line 76 to 89 as “Furthermore, regarding the operational accuracy of fully autonomous ships, the experiment results have shown that the collision avoidance routes made by fully autonomous ships are likely to be human-level [11].”

  1. Figure 5 (page 7) shows route optimisation. The result can be true for static objects, but will be incorrect for moving objects. From the text it is not clear what kind of objects they are.

Authors’ Response: Thank you very much for the comment. Figure 5 is to demonstrate the LCD model. It is to our understanding that the LCD model demonstration will be more understandable with static objects. Even though PAD and DPAD is based on the condition that own ship is the give-way vessel which takes collision avoidance actions by changing her course, and the target ship is the stand-on vessel which keeps her course and speed, the LCD model will also apply on the dynamic DPADs according to the rapid updating speed of AIS dynamic information if own ship change her speed or target ship change her speed and course. The relevant context have been added between line 243 to 244 and line 247.

  1. Figure 7 (page 9) is not clear. Distances, vectors and lines cannot be distinguished. All black lines may be correct, but the colored lines raise several questions. The text does not resolve this. The same for figure 8. As a result, I cannot verify if the conclusions are correct.

Authors’ Response: Thank you very much for the comment. Figures 7 to 11 have been altered with colors, and the context has been added to the concerning paragraphs for each figure. We would kindly suggest the reviewer to read the concerning part and conclusions again.

  1. The calculations on page 11 to 15 are very different from the approach I am used to. Because it was hard to follow, I did not verify if all is correct.

Authors’ Response: Thank you very much for the comment. Since the color and context for figure 7 to 11 has been added. The computational approach should be understood more easily. Moreover, we have done our best to present the computational process despite the fact that MCARP model might be relatively intricate, and we would kindly suggest the reviewer to read the concerning part again.

  1. Page 16 starts describing a simulator experiment. In table 4, the position of T3 is incorrect. The navigator of O maintains a minimum CPA of .675NM. The ships T4 and T5 pass at a smaller distance. Manoeuvring of the T-ships can be expected, possibly changing the entire outcome. Compare this also with the claim about superior automation vs. humans.

Authors’ Response: Thank you very much for the comment. We have modified the latitude of T3 to 24°59.00’ N to correct the geographical location of ship T3 in table 4. Moreover, the simulation results were output based on the collision avoidance routes which avoid DPADs. As specified in the section of the geometric model of DPAD, the starboard side of DPAD (which is on the starboard side of target ship) will have a length of r+0.14 nm, whereas the port side of DPAD (which is on the port side of target ship) will have a length of r-0.14 nm. Therefore, if the preset DCPA is 0.675 nm, which is r in the model, the collision avoidance distances while passing target ships might be between 0.53 nm to 0.81 nm.  According to the results, no passing distance is less than 0.53 nm. Some of the passing distance are larger than 0.81 nm, which is theoretically and practically acceptable. It has been stated in line 484 that all T-ships are stand-on vessels.  From the simulation results, it is likely to expect that the collision avoidance routes made by MCARP is human-level. The relevant context have been added between line 541 to 543 and line 559.

Reviewer 3 Report

I think it is an interesting study with some contribution to the field.

The abstract is a bit strangely shaped. It is too lengthy, has too large introduction.

The paper generally makes of a lot of strange and unsupported claims. For instance in the abstract the authors state the information can be provided to MASS through graphics overlay.

Claims that human errors contribute to 80% have been overturned since time ago by the relevant research. The authors need to update their claims. 

An abbreviation and nomenclature list would be useful in the paper.

Literature review could be enhanced with more references. Your unique contribution needs to be more clearly stated.

Language require improvements.

Author Response

  1. The abstract is a bit strangely shaped. It is too lengthy, has too large introduction.

Authors’ Response: Thank you very much for the comment and support. The abstract has been revised accordingly by omitting the excessive introduction.

  1. The paper generally makes of a lot of strange and unsupported claims. For instance, in the abstract the authors state the information can be provided to MASS through graphics overlay.

Authors’ Response: Thank you very much for the valuable comment. We have rewritten the concerning context from line 20 to 24 (“By operating the MCARP on ArcGIS, DPADs and several effective collision avoidance routes can be generated and be imported into ECDIS based on AIS information in large scales and ample time. The graphics overlay of DPADs and effective routes on ECDIS can serve as the collision avoidance strategy references for the personnel controlling the Maritime Autonomous Surface Ships.”) to make it more understandable.

  1. Claims that human errors contribute to 80% have been overturned since time ago by the relevant research. The authors need to update their claims.

Authors’ Response: Thank you very much for the comment. It should be around 60 to 90% according to the vessel type. We have revised the context and updated the citation between line 35 to 37.

  1. An abbreviation and nomenclature list would be useful in the paper.

Authors’ Response: Thank you very much for the comment, the nomenclatures have been listed in table 2.

  1. Literature review could be enhanced with more references.

Authors’ Response: Thank you very much for the comment. The relevant context has been added between line 175 to 194. The reference quantities of the article have also been increased to 33.

  1. Your unique contribution needs to be more clearly stated.

Authors’ Response: Thank you very much for the comment. The potential research gaps and the unique contribution has been added or revised between line 195 to 208.

Reviewer 4 Report

Dear editors and authors

This work considers the problem of obstacle avoidance collision in marine unammned and supervised vehicles.

The problem is of course very itneresting.

Although the introduction and problem description is very well presented, the problem for me is poorly addressed. The methodology is not rigorous, and to an extent simply feels like it is sort of example based approach. It is not systematic, and hencr not rigorous and for me not really implementable.

--No real algorithm is rpesented to solve this problem. Path planning has a vast literature, and the problem must be addressed in a systematic matter, providing a step by step algorithmic approach. This is not performed here. 

--Due to the above issue, for me implementability by readers, which is essential in all publications, is really not possible.

--Moreover, I did not get any real proof of concept for the method presented.

Due to the above, my suggestion would be rejection of the work, and a major rewriting from the authors perspective.

Some more comments can be found below

--line 192, LCD must be defined earlier as it is already mentioned as initials.

-line 212, note route 1 in green in the 1st graph.

--lines 220-224, and figure 1. The description is good, but it can be improved. For this, I would suggest using gray to denote the areas under study each time.

--check (5), p is not equal to d in the figure.

--figure 7 and subsequent ones, too may lines, maybe use colour, or maybe break it into subfigures.

some proofread is required.

Author Response

  1. No real algorithm is presented to solve this problem. Path planning has a vast literature, and the problem must be addressed in a systematic matter, providing a step-by-step algorithmic approach. This is not performed here.

Authors’ Response: Thank you very much for the valuable comment and your support. This study proposed a collision avoidance model as a “route planning concept” for MASS. The MCARP model, which includes the LCD model, could be integrated into ECDIS as a module. By the quick and accurate renewal nature of AIS information captured and displayed on ECDIS, MCARP model is likely to be operated in a larger scale within 2 to 10 seconds. Future studies can consider about extending this piece of work by applying renowned route planning method such as Ant Colony Algorithm, Particle Swarm Algorithm, and A-star Algorithm. We have added this detailed statement in “Suggestions for future works” section between line 577 to 582 to make the future research directions more understandable.

  1. line 192, LCD must be defined earlier as it is already mentioned as initials.

Authors’ Response: Thank you very much for the comment. LCD has been revised to be defined in full form in line 215.

  1. line 212, note route 1 in green in the 1st graph.

Authors’ Response: Thank you very much for the comment. We have rewritten the context between line 251 and 254 as “After analyzing the obstacle avoidance route on the starboard side in Fig. 5(a), the optional obstacle avoidance route 1 connecting waypoints S, A5, and E is constructed, and then the port side obstacle avoidance route will be evaluated”. This revision will make the reader understand which route is being discussed.

  1. lines 220-224, and figure 1. The description is good, but it can be improved. For this, I would suggest using gray to denote the areas under study each time.

Authors’ Response: Thank you very much for the comment. Graphs figure 5(a) to 5(e) have been altered with shallow blue areas to represent the areas under study.

  1. check (5), p is not equal to d in the figure.

Authors’ Response: Thank you very much for the comment. According to figure 6, it can be seen that p = d* sin (C). If the ship adopts parallel sailing, course as Cn = 90° (due east) or 270° (due west). C as course angle will be 90°, and sin (C) is thus 1, which will create the equation (5).

  1. figure 7 and subsequent ones, too may lines, maybe use colour, or maybe break it into subfigures.

Authors’ Response: Thank you very much for the valuable comment. Graphs 7 to 11 have been altered with colors, and the context has been added to the concerning paragraphs for each figure.

Round 2

Reviewer 2 Report

All my issues have been adequately addressed

Reviewer 4 Report

Dear editors and authors

Although the approach is mathematically less rigorous, I can understand that it may be more appropriately described for the audience of the journal. Moreover, the authors have well addressed my comments.